# In Vitro Dissolution and Permeability Testing of Inhalation Products: Challenges and Advances

**DOI:** 10.3390/pharmaceutics15030983

**Published:** 2023-03-18

**Authors:** Ali Nokhodchi, Salonee Chavan, Taravat Ghafourian

**Affiliations:** 1Lupin Inhalation Research Center, Lupin Pharmaceuticals, 4006 NW 124th Ave., Coral Springs, FL 33065, USA; 2Pharmaceutics Research Laboratory, School of Life Sciences, University of Sussex, Brighton BN1 9QJ, UK; 3Department of Pharmaceutics and Drug Delivery, School of Pharmacy, The University of Mississippi, Oxford, MS 38677, USA; 4College of Pharmacy, Nova Southeastern University, Fort Lauderdale, FL 33328, USA

**Keywords:** orally inhaled drug products, dissolution test, fine particle fraction, particle collection, impactors

## Abstract

In vitro dissolution and permeability testing aid the simulation of the in vivo behavior of inhalation drug products. Although the regulatory bodies have specific guidelines for the dissolution of orally administered dosage forms (e.g., tablets and capsules), this is not the case for orally inhaled formulations, as there is no commonly accepted test for assessing their dissolution pattern. Up until a few years ago, there was no consensus that assessing the dissolution of orally inhaled drugs is a key factor in the assessment of orally inhaled products. With the advancement of research in the field of dissolution methods for orally inhaled products and a focus on systemic delivery of new, poorly water-soluble drugs at higher therapeutic doses, an evaluation of dissolution kinetics is proving crucial. Dissolution and permeability testing can determine the differences between the developed formulations and the innovator’s formulations and serve as a useful tool in correlating in vitro and in vivo studies. The current review highlights recent advances in the dissolution and permeability testing of inhalation products and their limitations, including recent cell-based technology. Although a few new dissolution and permeability testing methods have been established that have varying degrees of complexity, none have emerged as the standard method of choice. The review discusses the challenges of establishing methods that can closely simulate the in vivo absorption of drugs. It provides practical insights into method development for various dissolution testing scenarios and challenges with dose collection and particle deposition from inhalation devices for dissolution tests. Furthermore, dissolution kinetic models and statistical tests to compare the dissolution profiles of test and reference products are discussed.

## 1. Introduction

Over the years, orally inhaled drug products (OIDPs) have been created to treat lung diseases locally. This method of administration is also presently being researched for the systemic distribution of medications because the lungs offer a significant absorption region and can bypass first-pass metabolism. Exubera, an insulin inhalation formulation sold in the United States and Europe, was the first commercially available formulation with this scope. Although this drug was withdrawn only after one year for business reasons [1], the use of OIDP to treat systemic illnesses is still of great potential benefit to patients and a focus area for the scientific community and pharmaceutical corporations. When compared to parenteral delivery methods, there is no need for needles, and patient compliance is higher [2]. The administration of OIDPs relies on the use of devices including nebulizers, pressurized metered dose inhalers (pMDIs), or dry powder inhalers (DPIs), each suitable for delivery of a different formulation type. The difficulties with these formulations are in controlling both the devices’ performance and the manufactured formulations’ physicochemical qualities. The establishment of efficacy and bioequivalence for OIDPs is considerably more difficult than that necessary for oral dosage forms due to the nature of the administration site and the complexity of the delivery of OIDPs.

The process of solid materials being dissolved in a solvent to produce a solution is known as dissolution. The affinity between the solid substance and the solvent, as well as the solid’s crystal packing factors, govern the process [3]. Although performing the in vitro dissolution test for solid oral dosage forms is a common practice recommended by pharmacopoeias to ensure quality control of the dosage forms, this has not been the case for OIDPs such as dry powder inhalations. Instead, traditionally only the delivery of the active pharmaceutical ingredient (API) from the device and drug deposition in the lung were considered important when developing OIDPs [4,5,6]. This is due to API deposition being often the most complicated step of pulmonary drug delivery, especially with the small dose, locally-acting OIDPs. It is blatant that only ideal-sized drug particles can deposit efficiently in the lung, and particles larger than 6 µm have less chance to enter the lung for deposition. The small drug particle size also helps increase the rate of dissolution of the drug deposited in the airways. Because of the above-mentioned reasons, the results on delivered dose and aerodynamic particle size distribution rather than the drug dissolution profile have been considered by the regulators. Deposition testing involves the administration of a particular API from a designated delivery device and utilizing a pharmaceutical impactor/impinger to determine the deposited drug content. This simulates the actual dose delivered to the action site of the lung and is probably the most crucial stage for in vitro performance testing for inhalation pharmaceuticals.

The dissolution of (solid) APIs is a necessary step before their absorption can occur via epithelial cells in the respiratory tract. It seems dissolution is especially important for poorly water-soluble drugs, as the absorption of highly soluble drugs is less restricted by dissolution and their therapeutic effects are not remarkably affected by the presence of other materials in the formulations. On the other hand, with poorly water-soluble drug powders, the slow dissolution rate in the airways may be limiting the absorption. Slow-dissolving drugs are also anticipated to be more susceptible to mucociliary clearance of drug particles [7]. For these drugs, dissolution is more significantly reliant on the type of excipients used [6].

In 2008, the inhalation Ad Hoc Advisory Panel of the USP decided that there was no concrete evidence to show that the dissolution of orally inhaled drugs was kinetically and/or clinically vital for the existing DPI formulations on the market [6]. Although performing the dissolution test for DPIs with a small particle size and a very small dose was not considered to be critical, with the current trends in the market moving towards high-dose and poorly water-soluble drugs, the dissolution test is becoming increasingly important [8].

Dissolution testing in vitro allows for the distinction between the efficiency of different formulation types and provides an approximation of how drugs would dissolve in vivo. In addition, it is frequently employed in quality control (QC) projects, including batch-to-batch uniformity and stability assessments, and the identification of manufacturing errors [6]. Depending on the purpose of the orally inhaled drugs, different dissolution profiles would be desired. For example, for a systemic effect, a formulation that allows for fast dissolution and high permeability may often be desirable to achieve a quick onset of drug action. On the other hand, for a long-acting local effect, a formulation with a low dissolution rate may be more desirable, helping to achieve a long duration of action, such as in the long-acting beta-adrenergic medicine olodaterol or other highly soluble medicines such as tiotropium bromide. Generally, inhaled corticosteroids, such as budesonide, fluticasone propionate, and fluticasone furoate, show low dissolution rates and hence s long residence time (2–7 h) in the lung [9,10]. The dissolution of an inhaled drug powder is controlled by inherent factors, such as the physiological nature of the deposition site and the physicochemical properties of the drug, as well as many modifiable parameters that can be adjusted. A summary of all the important factors is listed in Figure 1.

Compared with animal models, in vitro methods for assessing OIDPs offer mechanistic insight into the process of pulmonary drug bioavailability, as well as being advantageous in terms of cost-effectiveness, reproducibility, and accuracy of predictions. Contrary to oral dosage forms with standard dissolution and permeability testing methods, the in vitro methods for dissolution and membrane permeation of orally inhaled particles are challenging, and there is a lack of standardized testing methods for these formulations. This is due to the additional complexity arising from the reproducibility and appropriateness of methods for the collection of respirable particle doses (e.g., from cascade impactors) and the difficulty of designing a reliable dissolution and/or permeability method. A further consideration unique to inhalation products is the epithelial permeability occurring at the air/epithelium interface rather than the liquid/epithelium interface in gastrointestinal absorption, adding to the difficulty of simulating the process. This review will provide an overview of all the in vitro dissolution and permeability testing methods and recent developments in the technology. In vitro methods have seen some major advancements in recent years with technological developments such as 3D cell cultures, 3D printing of tissues, and organ-on-a-chip microfluidic devices, which are heading to replace animal testing in the near future. The potential applications of such advanced technology in in vitro testing of inhalation products and some directions for future research have been discussed, including aerosol particle collection, instrumentation, and optimization of dissolution medium for inhaled drug particles.

## 2. Process Overview of Dissolution and Permeability Testing for Inhaled Products

There are several factors to consider during the design of a dissolution or permeability testing method for inhaled drug powders. These factors have been summarized in Figure 2. The dose deposition device, dose collection method, dissolution methodologies, and choice and volume of the dissolution medium are essential factors to consider for carrying out these experiments and evaluating the results [11]. Based on Figure 2, based on the sample type, first, a suitable method should be adopted for the collection of drug particles (mainly less than 5 μm in size). This should closely mimic the in vivo deposition of fine drug particles in the airways and is a crucial step for the in vitro simulation of drug dissolution and absorption through the lungs. After dose collection, the dissolution and/or permeability method should be chosen, followed by the conditions of the experiment, such as the volume and composition of the dissolution medium. A suitable analytical method is also important to detect and determine the amount of drug dissolved or permeated. More details about these have been described in other sections of this review article.

It will be noted that, due to technical restrictions, some in vitro dissolution testing methods may be very different from the physiological condition in the airways, where there is a thin layer of highly viscous mucus present on a large membrane surface area, which provides different levels of membrane permeability depending on the nature of the drug and mucociliary clearance. Nevertheless, a measure of the dissolution rate can still contribute to improving the prediction accuracy of in vitro-in vivo correlations. The goal of various designs of dissolution testing and permeability assessments is to best simulate the process of drug absorption from the lungs while accommodating various ranges of drug particles, including variabilities in particle size, crystal form, formulation, and drugs’ physicochemical properties.

## 3. In Vitro Dissolution and Permeability Testing Methods

At present, there is no fixed and standardized dissolution technique accepted by regulators and pharmacopoeias. With growing evidence from studies showing the importance of the dissolution of OIDPs for better in vitro to in vivo correlations and the increasing use of this in the pharmaceutical industry, it is highly likely the dissolution testing of OIDPs will be implemented in pharmacopoeias in the future [12].

The efficient dissolution of inhaled drug particles is one of the key parameters that ensures better performance of OIDPs in the lung. In the last two decades, researchers have developed many dissolution methods that can be used to trace the dissolution of drug particle depositions. It is vital that the methodology can provide reproducibility and robustness and that the dissolution medium can simulate the physiological lumen and discriminate the effect of particle size on the dissolution rate in vivo. There are several crucial factors in dissolution testing that should be considered when developing a method, including agitation rate, flow rate, temperature, and the viscosity and composition of the dissolution medium [13]. In addition, the collection of the physiologically relevant drug particles, i.e., the fine particle dose (FPD), is a crucial prerequisite stage for both dissolution and permeability testing that will be discussed in Section 4. From the various dissolution testing apparatuses used in other dosage forms, only some may be suitable to be adapted for inhalation products. Researchers modify the existing dissolution apparatus or invent new ones to suit the specific needs of OIDPs. These include adjustments made in the paddle dissolution apparatus, Franz diffusion cell, and flow through cell to make them suitable for the dissolution testing of OIDPs. Below is a description of how each method has been adapted to be used in the dissolution of OIDPs. A summary of the use of these methods for fine particle dose collections of OIDPs as reported in previous literature has been presented in Table 1.

### 3.1. Paddle Dissolution Apparatus

The “Paddle over disc” method (USP apparatus V) was originally developed and approved by the FDA for transdermal dosage forms, but it has also been used for dissolution studies in OIDPs. The dose collected from DPIs on a membrane is placed in a cassette. The cassette is then placed in the USP 2 vessel, and the equipment is operated as illustrated in Figure 3. In this setup, drug release is controlled primarily by diffusion mechanisms. The core benefit of this method is that it makes use of a standard USP 2 apparatus that can be combined with various aerosol particle collection filters placed inside the vessel and various filter holders. The main issue with this system could be the presence of dead space between the insert and the bottom of the vessel, which can prevent the circulation of the dissolution medium, around the holder [25]. This can be solved by increasing the stirring rate of the paddle over the holder [15,16].

One of the critical steps in this method is dose collection, which is explained later in the article. The setup of the dissolution testing with respect to the process of collecting aerosol particles presents a challenge with this approach. Specifically, a porous filter is used to keep the particles on the collection surface (see details in Section 4). This holding filter might prevent wetting and thicken the diffusion layer. By employing a surfactant in the dissolution medium and tailoring the membrane’s size and composition, this impact can be reduced [25]. Another consideration is that the dissolution rate could be impacted by the aerosol particle collector’s orientation in the bath, adding variability if it cannot be positioned consistently throughout the experiments [25].

In this method, the volume of the dissolution medium may be modified to account for the two extreme scenarios of dissolution: high medium volumes to create sink conditions that simulate rapid absorption (high permeability drugs), and low dissolution medium volumes that simulate gradual absorption of low permeability drugs. The paddle-over-disk method has been utilized by researchers to distinguish different inhalation products [16]. Although this method has its own disadvantages, such as deviations from the real conditions in the lungs with high dissolution volumes and hydrodynamic conditions [14,26,27], a good correlation was established between the mean dissolution time and the in vivo mean absorption time for a range of corticosteroids [28].

### 3.2. Flow through Cell Apparatus (USP Apparatus IV)

The flow through cell dissolution testers are specialized USP 4 apparatuses modified to accommodate the dissolution testing of inhalation powders (Table 1). Most often, for dissolution testing with this apparatus, aerosol particles are collected using the particle filter method. USP 4 is comprised of a media reservoir (the dissolution medium) that is circulated through a dissolution cell by the use of a pump, while the filter containing drug particles is maintained in a filter holder within the dissolution cell. More specifically, dissolution cells are comprised of a filter holder with porous screens to contain the drug particles and support the filter, as shown in Figure 4. The medium is normally maintained at a constant temperature using a water bath and/or oven. Figure 5 shows an example of such a setup; here, an HPLC pump and an HPLC column oven have been adapted [4].

In this method, the mechanisms of dissolution are diffusion and convection through the filter. Permanent sink conditions and a diminished impact of membrane diffusion during dissolution tests have been reported as the potential benefits of this method compared with the paddle-over-disk methods. However, this is subject to the conditions of flow through the dissolution cell. For example, very low flow rates, especially given the small particle size of the inhalation drugs, can still lead to considerable drug concentration in the vicinity of the powders. Hence, it is advisable that the sink conditions be examined by measuring the API concentrations in the fractions collected during the dissolution testing [4]. In addition, the geometry of the filter holders may be impactful. The flat design of the dissolution cell may generate a strong fluid speed at the center but a diminishing flow gradient towards the perimeter, resulting in possible non-sink conditions arising locally at the periphery. Furthermore, it is important to note that the flow rate of the passing medium can potentially change the dissolution kinetic, which is another concern when flow through cell apparatus is used [4,11].

An additional risk with this method is the possibility of air bubbles between the two filters in the system. This can inhibit the wetting of drug particles, which can slow down the dissolution process due to the reduced surface area available for dissolution. Given the flexible nature of this method, the key to overcoming any of the pitfalls is to manipulate variables of the system, such as the flow rate, the membrane, and the geometry of the dissolution cell, to optimize the method for better mimicking the in vivo process of dissolution.

### 3.3. Diffusion-Controlled Cell Apparatus

As the amount of fluid in the lung lining is very low compared to the volume of fluid in the GI tract and as the fluid in the lung lining is more static, a suitable dissolution apparatus should be developed to reflect the in vivo conditions in the lungs. Based on these limitations, the Franz cell and Transwell^®^ systems were developed for testing the inhaled drugs.

Franz cells are widely used for diffusion experiments in assessing transdermal products, while Transwell^®^ has a variety of applications, including diffusion through membranes or passage through cell layers. For inhalation drugs, in these two systems, a small volume of donor phase is used to allow for the presence of an air-liquid interface, resulting in more biorelevant conditions for dissolution testing [14,26]. In this diffusion-controlled device, a membrane filter containing drug particles is inserted into a modified Franz cell (Figure 6) or a Transwell^®^ (Figure 7) system to conduct a dissolution test. In these methods, generally, a low volume of dissolution medium in the donor compartment is used, whereas the volume of the medium used in the acceptor compartment could vary between a few to 1000 mL.

With an agitated (such as a Franz cell) or non-agitated system (such as a Transwell^®^), this method seeks to more accurately simulate the in vivo environment. However, it is noteworthy that the measured effect is a sum of two distinct processes, namely dissolution and membrane diffusion, with diffusion expected to be the main driving mechanism [29] for most drugs and membrane pore sizes [30]. As a result, it might be quite challenging to discriminate between the dissolution rate and membrane diffusion effects if the study aims to identify mechanisms. Moreover, different membrane types/pore sizes will have a profound effect on the dissolution profile. In one study using budesonide and a proprietary drug, two membrane types, polyester and polycarbonate, interacted with the test material, impacting the integrity of the tests, while two other membranes, regenerated cellulose and Isopore PC, were found to be suitable for the dissolution testing [26].

In this method, each experiment should include a determination of the diffusion coefficient, tests for repeatability, and measurements of the adsorption of the substance to the membranes. This is necessary to obtain dissolution rates that are comparable between different experimental settings. When considering the in vivo dissolution of drug particles in the lung, even if the whole inhaled dose is not dissolved in the available amounts of the lung fluid lining, the systemic circulation can ensure a sink condition, at least for the highly permeable hydrophobic substances. In diffusion-controlled cells, a high diffusion coefficient through the membrane along with low membrane retention is necessary to accurately portray this in vivo state in the in vitro setting and to prevent an unrepresentative non-sink condition.

Respicell^TM^ is a new custom-built dissolution apparatus for inhaled drug particles, designed at the University of Parma, Italy, by Sonvico et al. in 2021 [22]. This apparatus is similar to Franz cells, with a design suitable for assessing the dissolution of the respirable dose deposited on the filter of a fast screening impactor (FSI). As shown in Figure 8, the Respicell has two compartments for holding the donor and receptor phases. The two compartments are connected to each other using a metal clamp, with a diffusion area of around 30.2 cm^2^ between the compartments for the insertion of the glass fiber filter that contains the respirable dose of the aerosolized product. An air-liquid dissolution condition is held in the donor compartment using a limited volume of medium [22].

### 3.4. Dissolvit^®^

Dissolvit^®^ is a more complex in vitro technique than the Franz cell and Transwell^®^ diffusion cells; in addition to a membrane, it includes a mucous gel layer simulant. This setup is expected to be more precise in simulating the dissolution and absorption of inhaled drug particles in the lung [21,31,32]. As shown in Figure 9, it consists of a single-use dissolution cell, a pump for the perfusion of the medium through the chamber, and an inverted microscope for optical tracing of the dissolution of particles. After passing through the dissolution cell, the medium is collected in a fraction collector. The dissolution cell has a reversed set-up, where the glass cover containing drug particles is placed at the bottom, in immediate contact with a mucous-simulant layer glued to a porous polycarbonate membrane at the top. The medium flows in the chamber at the top of the membrane. In this system, the physiological environment, i.e., the air-blood barrier, mucus, blood flow, and permeation from the lung epithelium to the bloodstream, has been taken into consideration. The technique was able to differentiate the dissolution profiles of two corticosteroids, fluticasone propionate and budesonide [21].

Gerde et al. further investigated this method by comparing it with the ex-vivo dissolution and absorption kinetics derived from the isolated, perfused, and ventilated lung of the rat using two popular inhaled drugs, nicotine and fluticasone propionate [31]. The dissolution and absorption of the lipophilic drug fluticasone propionate were slower in Dissolvit^®^ than in the isolated lung, while for nicotine, similar kinetics were observed [21]. This led to the conclusion that it is mainly lipophilic drugs (log p > 0) that benefit most from the use of sophisticated in vitro systems, such as Dissolvit^®^, that mimic the morphometry and physiological properties of the airways. This system was recently successful in securing US FDA support for further development and method validation as a potential ‘golden standard’ for dissolution testing of inhaled drugs [33].

### 3.5. Cell-Based Permeability Assays

Methods that are able to deposit the drug particles onto cell culture monolayers could potentially offer a more accurate in vitro model [23,24] of both dissolution and absorption in vivo while also offering simplicity and high-throughput potential as compared with ex vivo and in vivo methods. The cell-based methods may overcome the shortcomings of the dissolution methods that are not able to factor in the effect of the mucus layer and lipophilic membrane permeability in enhancing the dissolution and absorption of poorly water-soluble drugs [34]. The use of permeability values from cell cultures such as Caco-2 and MDCK is a common practice for the estimation of intestinal absorption of drugs [35]. Such permeability measures have been used in the biopharmaceutics classification system (BCS), which guides the FDA’s waivers of certain in vivo studies for immediate release, solid oral dosage forms, and suspensions [36]. The application of cell-based models for inhalation products has seen a rise in popularity in recent years, and there have been calls to develop similar BCS for OIDPs [37]. A proposed framework for inhalation-based BCS has been outlined recently. Similar to BCS for oral dosage forms, the framework incorporates two in vitro properties of the drug molecule, solubility and permeability (measurable from cell-based methods), but in addition, it incorporates two drug formulation properties specific to OIDPs, namely dose and dissolution [38].

Examples of studies that incorporate cell-based models for orally inhaled formulations are listed in Table 1. Cell-based methods rely on primary cells or immortal cell lines from the respiratory epithelial origin (such as Calu-3) grown on filter supports, ideally preserving the intercellular tight junctions and polarized features [39]. Various cell lines are available from bronchial and alveolar epithelia for permeability or toxicology studies, or to model disease states. Some of the most widely used cell lines for permeability studies are Calu-3 cells from the bronchial epithelium and A549 cells of alveolar origin. A main feature in modeling respiratory epithelium using cell culture is the necessity for the air-liquid interface at the apical side of the monolayer. Hence, cells are grown on a membrane with the medium on one side of the membrane, i.e., the basolateral side, and the apical side exposed to air, often with oxygen concentrations [40]. This setup will facilitate polarized differentiation of the cells, formation of intercellular junctions, and expression of native proteins, including metabolizing enzymes [41].

Several studies have shown a reasonable correlation between permeability values from cell-based models and the in vivo lung absorption rates in animals [42,43]. Other studies indicate that cell-based methods may fail to account for the effect of the mucus barrier (which is present in vivo) on increasing the absorption of poorly water-soluble drugs [34].

Alongside developments in cell culture technology, some research efforts have focused on dose deposition devices to be used with these (mono/multiple) cell layers [44,45]. Specific deposition devices have been developed for the uniform and reproducible deposition of particles on cell cultures and Transwell^®^ vessels, or perfused cell systems. Pharmaceutical Aerosol Deposition Devices on Cell Cultures (PADDOCC) [23], Vitrocell^®^ Powder Chamber [46], and custom-made devices [47,48] are examples of such systems. Recent developments in cell-based methods also include the integration of fine particle dose collection impactors with the cell culture component to allow easy application [49].

Efforts have been made to develop specialized cell cultures that improve the modeling of the permeability properties in comparison with established cells such as Calu-3. These include the use of new immortalization methods for alveolar cells [50], the generation of alveolar epithelial cells from induced pluripotent stem cells (hiPSCs) [51], or co-cultured cell types of various primary origins, e.g., alveolar epithelium and macrophages [52]. Most cell-based systems are static and use cells cultured on Transwell^®^. PerfuPul is a perfusable cell-based system designed for permeability studies that allow aerosol deposition on the air/liquid interface of Calu-3 cells [53]. A full discussion about the characteristics of various cell lines and their performance in modeling the permeation of OIDPs from the lungs can be found elsewhere [54].

### 3.6. Recent Advancements in Technology for In Vitro Testing

Benefitting from advancements in science and technology, numerous in vitro models have been developed to support pharmaceutical drug development as well as the safety assessment of consumer products and environmental pollutants. These in vitro systems aim to act as alternatives to animal testing, increase the experimental throughput, and present a more accurate model for clinical situations than animal models. Despite the availability of sophisticated systems such as 3D cell cultures, reconstructed tissues, and organs-on-a-chip, many of these are specialized for toxicity testing purposes rather than for use as pulmonary absorption models. Examples of tissue models that, according to their suppliers, have the barrier function and potential phenotype for drug delivery experiments are those developed commercially by Epithelix (MucilAir™ and SmallAir™) and MatTek 3D (EpiAirway™, EpiAirway-FT™, and EpiAlveolar™). A wider application of these can be seen in the literature for toxicity studies, while the applicability for drug delivery research is being investigated [40]. Furthermore, the authors of the current review article recommend referring to a recent review article published by Eedara et al. (2022) where more examples of the absorption of orally inhaled drugs can be found [55].

Another emerging field that may contribute to the future of in vitro pulmonary drug delivery testing is organ-on-a-chip technology. This technology has been made possible due to advancements in cell-culture techniques and biomaterial microfabrication. A review of the currently available lung-on-a-chip models indicates greater accuracy may be achieved with these models compared with traditional monolayer cell cultures [56]. With the advent of more complex systems, the efficiency of lung-on-a-chip or alveolus-on-a-chip models in lung permeability estimations will highly depend on the implementation of functional human cells within the system [57,58].

## 4. Aerosol Particle Collection

Fine Particle Dose, defined as particles with an aerodynamic diameter ≤ 5 μm, is an established pharmacopeial method for the in vitro assessment of dry powder inhalers. This particle size range is thought to be the dose that reaches the desirable site of deposition during respiration [59,60], with larger particles having a higher probability of being deposited in the oropharynx area. Therefore, when performing in vitro studies, it is advisable to conduct the dissolution test on aerosolized particles in the potentially respirable size range (usually less than 5 µm) so the in vitro results can be an accurate estimate of the in vivo drug delivery efficiency of the products. Despite this, some earlier in vitro investigations of the dissolution and permeability of DPIs have used APIs directly or powders collected in a way that exact particle sizing is difficult to establish.

Product particles are normally aerosolized using various impactor designs and then the desired particle size range is collected on filters or membranes, which are then transferred to dissolution or permeability apparatus such as USP II, USP IV, Franz cells, or Transwells (see examples from Table 1). Some powder production methods, e.g., spray drying, can produce particles with uniform size and geometry, allowing the direct use of a specific mass of the powder on various dissolution apparatuses [14]. In Franz cells, Transwell^®^ setups, and cell-based methods, a uniform and reproducible deposition of fine drug particles is a major consideration for the reproducibility of the results. In this case, specialized deposition devices may be employed. Many of the deposition devices do not allow particle size separation. Examples are Astra-type liquid impinger, customized for direct deposition of aerosolized particles onto the cell monolayers [36], and dry powder insufflators such as the DP-4 from Penn-Century^®^, which were originally designed for precise pulmonary administration of dry powders to laboratory animals. Alternative systems may allow size-dependent depositions, such as the widely used Astra-type multistage liquid impinger (MSLI) [61]. Other commercial examples of deposition devices are the PreciseInhale^®^ exposure system (XposeALI ^®^3D cell exposure module) and Vitrocell^®^ Powder Chamber, which separate particle sizes by their time of flight.

Some of the most widely used commercial impactors for aerodynamic particle separation are the Next Generation Impactor (NGI), Andersen Cascade Impactor (ACI), fast screening impactor (FSI), or multi-stage liquid impinger (MSLI). For dissolution studies, specific filters and/or membranes are used for the collection of particles from these impactors [62]. Thought must be given to the possibility that filters may have an impact on the aerodynamic flow profiles of the particles. Figure 10 shows a glass fiber filter located on stage 3 of an NGI.

When using the USP II apparatus for dissolution testing, membranes with deposited drug particles from impactors can be directly placed into the dissolution medium or be covered with an additional pre-soaked filter/membrane before being placed into the medium. In this case, a filter cassette is used to keep the filter in place (Figure 11). If particles are collected on a glass fiber filter, a membrane filter, such as a polycarbonate filter, can be placed over the glass fiber filter containing the API particles.

In an investigation aimed at developing a standardized dissolution testing strategy, Arora et al. [17] focused on the collection of defined particle sizes and respirable particles. They achieved this by using an Anderson cascade impactor (ACI) with polyvinylidene difluoride (PVDF) filters on stainless steel collecting plates placed on stages 4 and 2. Son et al. [15,25] used a modified NGI where a dissolution cup was assembled with the detachable impaction insert at stages 4 or 5, depending on the deposited load, but with a defined particle size range. Other researchers have mounted particulate filters on stage 3 of the NGI [35] to capture drug particles with specific particle sizes at their working flow rate (Figure 10). Eedara et al., also modified a twin-stage impinger to collect respirable size particles of anti-tubercular drugs to investigate the dissolution pattern of the collected particles [63].

When using the various particle collection and deposition methods, it is important to note the impact of the particle size distribution on the measured dissolution rates. Of note is the fact that particle separation in impactors is based on the aerodynamic particle size. In carrier-based dry powder formulations, the primary drug particle size (geometric diameter) drives the dissolution process, whereas the aerodynamic particle size, which is a function of agglomerate size, density, and aerodynamic properties, is responsible for the deposition in the lung. Franek et al., exhibited the effect of primary drug particle size (geometric size) on the dissolution of AZD5423 (obtained from AztraZeneca). They showed that drug particles with an average diameter of 3.1 µm have a much slower dissolution rate than particles with a 1.7 µm mean diameter [19]. In addition, the impact of aerodynamic size on dissolution rate has been established by comparing material collected from different stages of impactors [15,17,25].

In these investigations, it is also important to note that the amount of drug collected at each stage of the impactors may have an impact on dissolution rate measurements. The effect of drug mass (e.g., the number of actuations) on the dissolution behavior has been demonstrated for hydrocortisone particles collected from various stages of an NGI [25]. In this example, the dissolution rate of drug particles collected from stage 6 of NGI, i.e., smaller particles, was less sensitive to the mass of particles collected compared to larger particles. Due to the massive impact of the particle mass on the dissolution rate, it is ideal if the same amount of drug mass is collected when comparing various formulations. In addition, in order to avoid having a multilayer of particles on the filter and to prevent the formation of drug agglomerates and wetting issues [16,61,64], a single actuation may be preferable when collecting the particles. On the other hand, for the benefit of the accuracy of analytical determinations, especially when a high volume of the dissolution medium is used, more than one actuation may be required to increase the sensitivity of the determination of APIs, as in HPLC. For example, in a study to collect drug particles using reduced NGI (rNGI), 5 actuations were performed to collect drug particles to obtain a quantifiable amount [25].

Depending on the choice of the dissolution method, the results are affected by the membrane used to varying degrees. Cell-based methods such as Franz cell, Transwell^®^, and Dissolvit^®^ are highly dependent on the diffusion process, hence the anticipated impact of the membrane type and efforts to mimic in vivo pulmonary membrane properties. With USP II methods, especially if the goal of experiments is purely the dissolution mechanism (and not permeation), an optimum membrane pore size may be identified that does not hinder the diffusion of the dissolved drug [15]. It has been shown that the thickness of the membrane and the tortuosity and size of its pores regulate the diffusion of dissolution medium across the membrane, hence the dissolution rate. The widely used polycarbonate membranes have a homogeneous thickness of 6 μm, and a bubble-free pore size of 0.05 μm. They appear to have uniform cylindrical swelling-resistant channels for easy diffusion and unrestricted dissolution [15,65]. A detailed discussion about the effect of membranes, along with a list of various commercially available membranes, can be found elsewhere [62].

## 5. Optimization of Dissolution Medium for Orally Inhaled Drug Particles

The dissolution medium is an important factor that impacts drug solubility and, hence, the dissolution rate of drugs. Therefore, careful consideration should be given to the choice of dissolution media in terms of its composition, volume, and stirring rate during the in vitro assessment of OIDPs, especially due to the lack of standardized methods. Common APIs used in the formulation of orally inhaled products have a wide spectrum of solubility, with current trends in drug discovery moving towards high molecular weight and low solubility drugs [66]. Moreover, a relationship has been established between solubility, which is a thermodynamic property in nature, and dissolution rate, which is a kinetic property [17,67]. For an appropriate dissolution rate determination, in practice, the dissolution medium needs to be optimized based on the drug solubility, by using solubility enhancers for poor-solubility drugs [11]. The incorporation of a surfactant in the dissolution medium is commonly used for assessing the dissolution of poorly water-soluble compounds [25]. In this case, surfactants may help increase the wettability and solubility of these drugs, allowing a readily detectible rise in drug concentration as a function of time. Surfactants such as sodium lauryl sulfate, tween 80, phospholipids, or other solubilizing agents such as alcohols also help preserve the sink conditions during the dissolution process [68]. Moreover, the use of surfactants is rationalized by mimicking the influence of natural surfactants present in the lung fluid [25,62,69]. An example of research comparing the effects of surfactants in a dissolution medium reported the faster dissolution of a poorly water-soluble drug (budesonide) in the presence of polysorbate 80 (≈90% dissolved within 1 h) compared to the dissolution medium without any surfactant (≈55% in 1 h) and that with dipalmitoyl phosphatidylcholine (≈60% in 1 h) [25]. Here, the results indicated that polysorbate 80 offered better solubilization of the drug powder compared to dipalmitoylphosphatidylcholine. An interesting approach to the design of a dissolution medium was developed by Bhagwat et al., who used semi-mechanistic models of pulmonary absorption to identify a dissolution medium that achieves the rate of in vivo drug dissolution [68]. The dissolution medium tested here consisted of various concentrations of Tween 80.

A number of studies have focused on the exact composition and pH of the fluid in the lung, and it has been reported that the composition is different in various parts of the lungs but also varies between healthy and diseased lungs [70] and changes with aging and infections. Although the composition of fluid in the lung is very complex, several simple aqueous media, such as water with phosphate buffer containing electrolytes (at the physiological level) or protein and phospholipids, have been developed to be used as biorelevant dissolution media. Some well-known simulated lung fluids that are commonly used as dissolution media and a guide to the preparation procedures have been presented in a review article [71]. Gamble’s solution is a model for deep lung interstitial fluid with a pH of 7.4. Artificial lysosomal fluid is more acidic (pH = 4.5) and simulates the lung fluids in inflammatory conditions. Modifications to these have been suggested for mimicking the lung environment following the deposition of particulate environmental pollutants [72]. In addition, some pulmonary surfactant extracts have been used as dissolution media in the dissolution testing of OIDPs. Pulmonary surfactant extracts from animal sources are typically clinically used for exogenous surfactant replacement in various disease states, such as infantile respiratory distress syndrome [73]. Examples are Survanta^®^, Alveofact^®^, and Curosurf^®^, which are commercially accessible and have been used in in vitro dissolution investigations [69,74,75].

There is between 10 and 30 mL of aqueous fluid in the lung, including lung surfactants, whereas the dissolution volume in the conventional apparatus ranges from 2 to 1000 mL, often much higher than the amount of fluid in the lungs. To use a low volume of dissolution medium, it has been suggested that the agitation rate should be enhanced to obtain the same dissolution as the large volume of dissolution medium [76]. However, the high agitation rate of the dissolution medium is different from in vivo conditions for inhaled products.

Another factor that should be considered when evaluating the dissolution pattern of orally inhaled drugs is the effect of excipients used in the formulations of MDI or DPI. However, it seems the influence of excipients on the dissolution rate of drugs could be minimized in experimental settings that incorporate dose collection from DPI formulations. Here, the particle size of the lactose carrier is too large to enter Stage 3 of NGI when capturing the drug particles for a dissolution test. Despite this, some DPI formulations incorporate lactose fines (<10 µm) to boost the performance of DPI formulations [77,78].

In this case, the API may show faster dissolution due to the instant dissolution of the lactose fines from API-lactose agglomerates in the dissolution medium. It should be noted that the agglomerates reduce the specific surface area, hence slowing down the dissolution, unless the agglomerates undergo deagglomeration when in contact with the dissolution medium [28].

The Influence of excipients on the efficiency of orally inhaled drugs is more pronounced in pMDI than DPI, as a wider variety of excipients are used in PMD formulations. For instance, it has been shown that an increase in the content of ethanol in solution-based MDI can alter the aerosol droplet size and the evaporation rate. Furthermore, the changes in the evaporation rate can change the morphology of particles after the evaporation, resulting in a notable alteration in the dissolution of the drug in a solution-based pMDI [79,80]. In another study, the use of glycerol in pMDI solution-based formulation restricted the access of water to lipophilic drugs, which can slow down the dissolution rate of APIs [81].

As an example of a suitable dissolution medium for a poorly water-soluble drug, Al Ayoubi et al. comprehensively studied the dissolution behavior of budesonide in diverse dissolution media in order to correlate the in vitro data to the in vivo data [82]. They used a modified twin-stage impinger to collect drug particles and suggested using only two actuations so a thin powder bed could be produced on the collection disk. This would reduce the impact of powder aggregation by reducing the surface area available for the wetting and dissolution of the drug [83]. Three media were used to study the dissolution behavior of budesonide (BD): Gamble’s solution, 0.2 M phosphate buffer (PB), and phosphate buffer containing 0.2% *w*/*v* polysorbate 80 (PBS). Gamble’s solution is a simulated lung fluid frequently utilized as a model for lung fluid even though it lacks surfactant [84,85]. The findings demonstrated that a significant amount of budesonide dissolved in these media within the 1st hour. The dissolution values after 2 h increased to 80.8, 78.9, and 95.8% for Gamble’s solution, PB, and PBS, respectively. These data were in agreement with the solubility data [83,86] in these media (14, 16, and 53 µg/mL in Gamble’s solution, PB, and PBS, respectively). The data indicates the role of the surfactant in enhancing the dissolution rate as well as the solubility of budesonide. Hence, surfactants (as in PBS) seem to provide a suitable dissolution medium for testing the quality of hydrophobic drugs by increasing the wettability and drug saturation solubility and speeding up the dissolution [11,87,88].

## 6. Application of Statistics to Dissolution Data

Models are used to mathematically describe the dissolution profiles of OIDPs and allow a quantitative comparison of the dissolution profiles when examining data obtained from dissolution tests. This is useful, for example, when aiming to make statements about the similarity or dissimilarity of dissolution profiles. Although statistics are frequently used in the dissolution assessment of many OIDPs, the actual details of the statistical methodology and the data structure are not always clearly discussed, which makes it difficult to accurately assess the quality of the results and the validity of the conclusions. To compare two dissolution profiles, two approaches have been employed. One is comparing the mechanism of drug release by fitting various kinetic models into the dissolution data. Common kinetic models include zero-order release, first-order release, Higuchi, the Hixon-Crowell equation, Korsmeyer-Peppas, and Weibull. The second approach is the statistical comparison of two dissolution profiles by computing the difference factor (*f*_1_) and similarity factor (*f*_2_). Both approaches are briefly discussed below.

### 6.1. Modeling of Dissolution Profiles

Generally, statistical models based on mathematical curves or release kinetic functions can be used to characterize dissolution profiles. These methods simulate individual dissolution profiles but do not offer a comparison of profiles that may be measured directly.

Based on the kinetics of drug release and the identification of the release mechanisms, dissolution curves can be described using model-dependent kinetic functions, such as zero-order, first-order, Higuchi, Hixon-Crowell, and Peppas models. When different kinetic models are used, generally the coefficient of determination (R^2^) is investigated to select the model that can best fit the dissolution data. However, in some cases, other parameters such as the sum of square residuals, the mean square error, and the Akaike information criterion are also considered [89,90]. The model fitting is often performed without consideration of the physicochemical properties of the drug or the expected mechanisms of its release. The various kinetic models relate the cumulative amount of the dissolved drug (y-variable) to the time as the x-variable. On the other hand, the dissolution kinetics can be mathematically modeled based on the theoretical description of the process of dissolution using Noyes-Whitney, Nernst Brunner, or derived models such as Hixon-Crowell to describe the diffusion process. However, in practice, the dissolution of drugs from OIDP formulations often does not follow the ideal diffusion process described by these theoretical diffusion models. This is the main reason for using semi-empirical or empirical models instead of theoretical ones. Some of the commonly used kinetic models are listed below:Zero-order release model: Q_t_ = Q_0(dissolved)_ + K_0_·t
First-order release: Q_t_ = Q_0(undissolved)_ (e^−k^_1_·t)
Higuchi model: Q_t_ = Q_0_K_H_·t^0.5^
Korsemey-Peppas (power law): Q_t_ = K_KP_·t^n^
Weibull model: log[ln(1 − Q_t_)] = b·log (t − T_i_) − log a

In the above kinetic equations, Q_t_ is the cumulative amount, fraction, or percentage of the drug dissolved at time t, and Q_0_ is the initial amount of the drug. In the zero-order release model, Q_0(dissolved)_ is the already-dissolved amount in the dissolution medium at time 0, while in the first-order release model, it is the full amount of drug that existed in the formulation that will be delivered to the dissolution medium (Q_0(undissolved)_). K_0_, K_1_, K_H_, and K_KP_ are the rate constants of these models. In the Weibull model, T_i_ denotes the lag time before the onset of the dissolution process, which may be zero in most OIDP cases, and a and b are model constants. In the Korsemey-Peppas model, n is the exponent parameter that indicates the mechanism of drug release.

Generally, poorly water-soluble APIs may show zero-order release kinetics. An example of this has been shown for orally inhaled drugs when the flow through cell method was employed to assess the dissolution rate of fiber filter-containing drugs [11]. Pseudo-zero-order release was also reported for fluticasone propionate dissolution in Transwell [17].

An example of a first-order release was reported for the dissolution of budesonide when Franz cells and USP apparatus number II were used to investigate its dissolution profile. However, when the flow through cell method was used, the dissolution data did not fit well with the zero-order model [16].

In the case of a controlled-release formulation for inhalation use developed by Salam et al., the dissolution rate was governed by the Higuchi model [14]. This formulation consisted of co-spray dried drug-polymer microparticles of disodium cromoglycate, which were tested by three different methods, namely USP apparatus II, flow through, and Franz cells. The fit of the dissolution data into the Higuchi model indicated that wetting and diffusion controlled the dissolution rate of the drug in these settings.

The power law model has been reported for paclitaxel and doxorubicin OIDPs [91]. In this research work, the dissolution of nano-in-micro formulations for DPI was investigated by suspending the formulation particles in 10 mL of buffer solution. The *n* values for various dissolution profiles in various buffers were above 0.2, indicating a non-Fickian dissolution.

The Weibull model, which assumes a linear plot between the logarithm of the dissolved amount of drug versus the logarithm of time, has been widely used for OIDPs. For example, Bhagwat et al. employed this to fit the dissolution data for corticosteroids obtained from a modified Transwell^®^ system [92]. In addition, the dissolution of various inhaled APIs (e.g., budesonide, fluticasone propionate) followed the Weibull model when the USP paddle apparatus and Transwell^®^ system were used [19]. The Weibull model was also employed by Hassoun et al. when they used the Dissolvit^®^ method to study the dissolution of OIDPs containing fluticasone propionate [93]. Despite its widespread use, a lack of a kinetic foundation and mechanistic insight into the dissolution process has been noted for this model by some authors [89]. However, Papadopoulou et al. performed a new comprehensive analysis of various dissolution data and suggested a correlation between constant b of the Weibull model and the constant n of the power model [94], indicating an indirect mechanistic conclusion may be drawn from b values regarding the dissolution process.

### 6.2. Methods for Comparing Dissolution Profiles

For oral solid dosage forms, a variety of techniques are frequently utilized for the comparison of dissolution characteristics. The most widely used methods are model-independent methods known as the difference factor (*f*_1_) and similarity factor (*f*_2_), developed by Moore and Flanner [95] and recommended by the FDA [96] for comparing two dissolution profiles between test and reference products.

Tsong and Hammerstrom [97] offer statistical techniques for solid oral products based on analysis of variance, either for a single dissolution time point (ANOVA or Student’s *t*-test) or for multiple dissolution time points (MANOVA). Even though multivariate methods (such as MANOVA) compare full dissolution profiles, they might not have enough statistical power to identify significant differences. ANOVA and *t*-tests only compare one dissolution time point; they might be more robust when a meaningful time during the course of the dissolution is picked for the comparison.

The comparison of the test (*T*) and reference (*R*) dissolution profiles is currently performed using two model-independent methods based on a difference factor (*f*_1_) and a similarity factor (*f*_2_):$$f_{1}=\frac{\sum_{j=1}^{n}\left|R_{j}-T_{j}\right|}{\sum_{j=1}^{n}R_{j}}\times 100$$
$$f_{2}=50\times \log\left\{{\left[1+\left(1∕n\right)\overset{n}{\underset{j=1}{\sum }}{\left|R_{j}-T_{j}\right|}^{2}\right]}^{-0.5}\times 100\right\}$$

The percent inaccuracy across all time points is measured by the difference factor, or *f*_1_, to see if there is evidence of a substantial difference between the two dissolution profiles. The similarity factor, *f*_2_, on the other hand, evaluates if there is enough proof of similarity between the two profiles using an equivalence approach based on mean squared differences. The following acceptance standards for oral solid dosage forms are recommended by both the FDA and EMA:If the *f*_1_ value is less than 15 (0–15), there is no difference (i.e., there is no proof of difference or there is no “signal over noise” to be seen).*f*_2_ values between 50 and 100 (above 50) suggest similarity (i.e., there is evidence that there is no important difference). Additionally, authorities (e.g., FDA) want that the sponsor compares dissolution profiles using the similarity factor using at least 12 distinct dosage units. When comparing dissolution profiles to support “biowaivers” for process scale-up or formulation adjustments for oral solid dosage forms, regulatory advice has mostly focused on the *f*_2_ measure. Generally speaking, “suitable statistical testing with the rationale” is permitted under the FDA guidance [98,99]. The EMA recommendation [100] suggests comparing distinct time points, model parameters, and similarity variables. The addition of several acceptance limits for various size ranges [25], which also apply to tests other than *f*_1_ and *f*_2_, is another factor for OIP particles fractionated by cascade impactors.

## 7. Conclusions and Future Perspectives

Parallel to technological advancements in cell-based methods, tissue engineering, and material fabrication techniques, the evaluation of the drug delivery efficiency of pulmonary drug delivery systems has evolved substantially. These techniques aim to eventually replace in vivo animal experiments, which can no longer be considered the gold standard in bioavailability, pharmacokinetics, or drug safety investigations. The lifting of legal requirements for animal testing of new drugs by the U.S. Food and Drug Administration (FDA), signed by President Joe Biden in December 2022, is a landmark change that is yet to be exploited by drug discovery and development disciplines utilizing the in vitro techniques discussed here as well as an in silico information-technology-driven approach. 

To be more aligned with in vivo conditions, new technology should be able to simulate the presence of a small amount of fluid in the lungs, mucociliary clearance, the presence of viscous mucus and its composition, along with adequate multi-cell diversity and sink conditions. Along with these complex cell-based techniques, such as microfluidic lung-on-chips, simpler in vitro experimental setups will continue to be used due to several advantages. First, these methods allow for detailed mechanistic studies since they focus on specific unit processes leading up to pulmonary drug absorption. The second advantage of these may be their higher throughput and cost-effectiveness, which makes them more accessible to specialized research and development entities. In addition, these less complex in vitro methods have fewer intervening experimental parameters, reducing the variability in experimental results. Moreover, a combination of various in vitro experiments may be employed to mathematically model the clinical bioavailability. This review is a compilation of various in vitro techniques that are used to simulate drug deposition, dissolution, and permeation from the lung epithelium. It has covered both the established techniques, with practical guidelines and experimental considerations, as well as upcoming novel techniques that may develop in the future for widespread use in product development research settings.

Following the drug deposition in the lungs, the dissolution of APIs is the first step before absorption occurs via epithelial cells in the respiratory tract. Due to the poor solubility of many of the orally inhaled drugs and the recent shift in drug discovery towards highly lipophilic, high molecular weight drugs with poor water solubility, dissolution kinetics is an extremely important process to be considered when assessing new formulations. It is also evident from the literature that dissolution studies may be less important for highly water-soluble drugs, whose dissolution and absorption from lung depositions are also expedited due to their fine particle size. Currently, performing the dissolution test for orally inhaled drugs is not required by the regulatory authorities due to the small dose (very small amount of API) of current OIDPs and their fine particle size in DPI formulation. However, with the market moving towards high-dose DPI formulations as well as an interest in DPI formulations for systemic delivery of a wider range of drugs that may be poorly water-soluble, this may change in the future. The established in vitro dissolution tests should have the capability to discriminate between different formulations and different APIs’ particle sizes. Although simple dissolution methods may not be able to closely mimic the exact in vivo conditions, they should be able to reproducibly differentiate various formulations with different performance efficiencies.

Dissolution techniques such as the paddle dissolution tester, flow through cell apparatus, or Franz cell are still popular for orally inhaled drugs. The selection of a dissolution methodology is critical, as the method of choice can impact the resulting dissolution kinetics. Methods differ not only in their instrumentation but also in the mechanism by which the dissolution process happens within these experiments. In methods such as Franz cells, where dissolution occurs mainly by diffusion, it is important to investigate the impact of membranes. Considerations should also be given to maintaining sink conditions since some methods, such as Transwell and Franz cells, employ only a low volume of dissolution medium, which will saturate quickly with low-solubility drugs. In these cases, a careful choice of dissolution medium composition becomes specifically critical to aid in maintaining a sink condition through the use of solubility enhancers as well as mimicking the natural surfactants present in the lungs.

When using the various particle collection and deposition methods, it is important to note the impact of the particle size distribution on the measured dissolution rates. It should be noted that particle separation in impactors based on the aerodynamic particle size may be affected by the presence of the filter used for the collection of particles. Hence, validation of particle collection methods is a crucial task required for robust results to be obtained. In addition, consideration should be given to the amount of the collected powder as well as a uniform powder distribution on the deposition membrane since these can affect the dissolution rate, especially in methods such as transwell^®^, Franz cell, and flow through. This is due to the variable wetting and agglomeration propensity of particles in high powder content areas of the membrane.

Finally, it is also important to explore what statistical tests are most suitable to compare the dissolution profiles of two inhaled products. This is important when inhaled generic products are being developed and the dissolution behaviour of the test product should be compared with the dissolution profile of the reference product. To overcome the hurdles and challenges associated with the dissolution of OIDPs, a good collaboration between academic institutions, the pharmaceutical industry, and regulatory bodies seems to be essential.

## Figures and Tables

**Figure 1 pharmaceutics-15-00983-f001:**
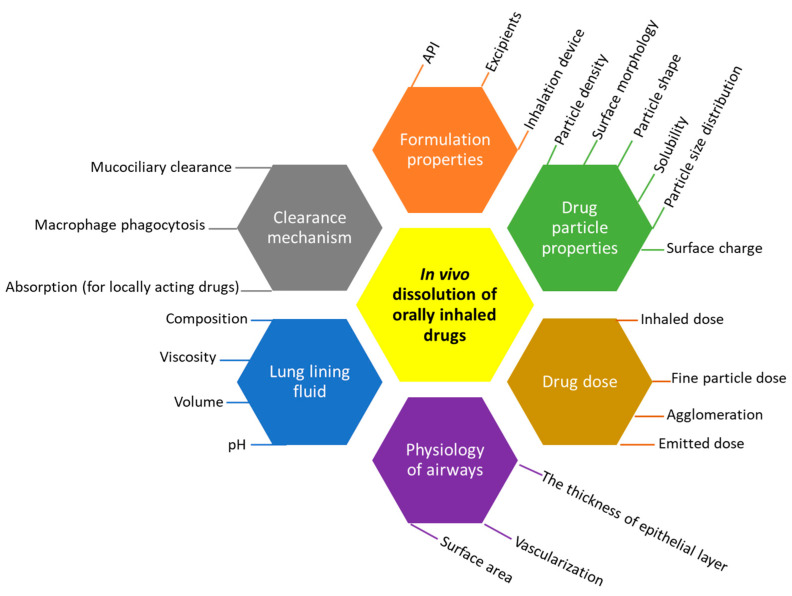
Parameters affecting the dissolution of inhaled API in the lung.

**Figure 2 pharmaceutics-15-00983-f002:**
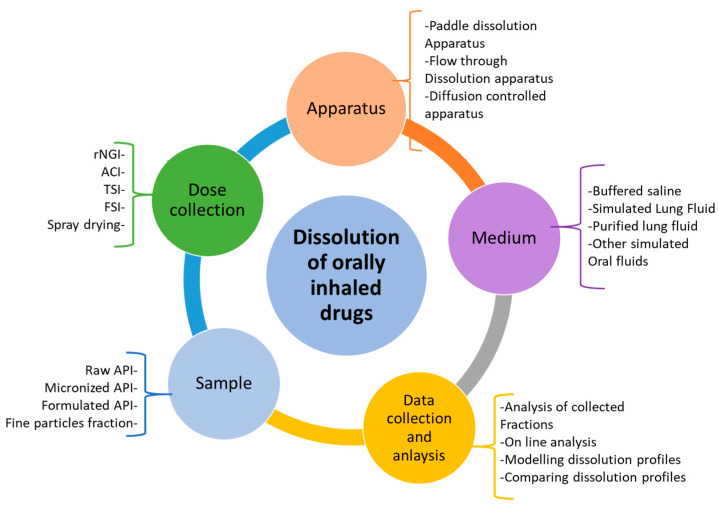
Potential stages of dissolution testing and parameters involved in each stage for the dissolution of inhaled compounds (rNGI = reduced next generation impactor; ACI = Anderson cascade impactor; TSI = twin stage impinger; FSI = fast screening impactor).

**Figure 3 pharmaceutics-15-00983-f003:**
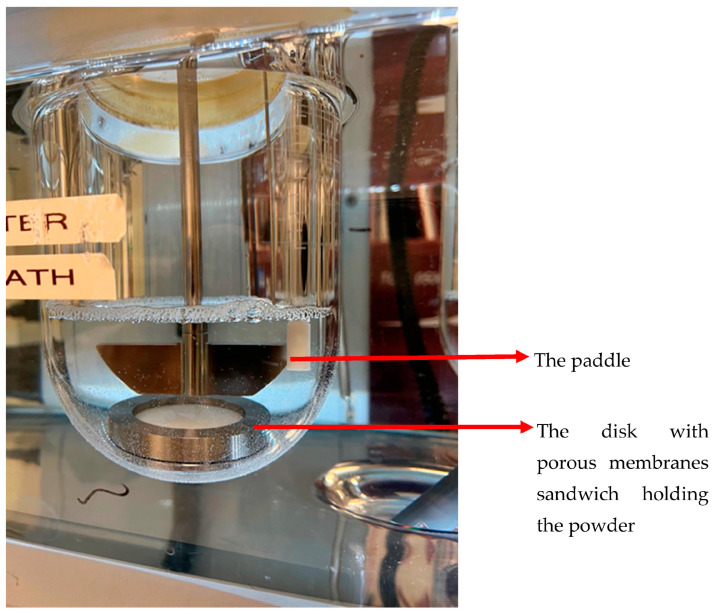
Paddle dissolution apparatus with insert holder and polycarbonate filter.

**Figure 4 pharmaceutics-15-00983-f004:**
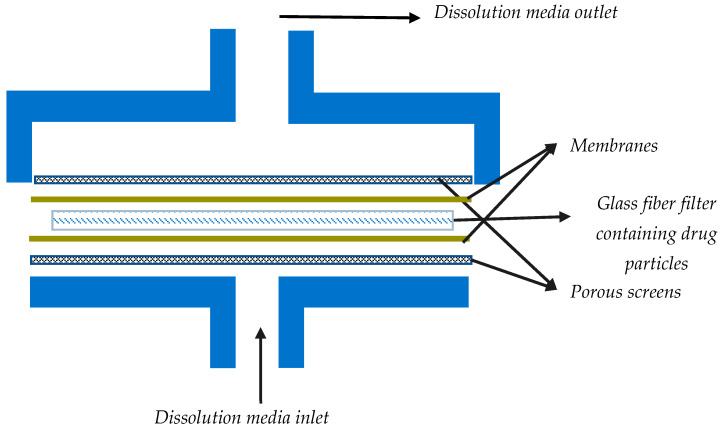
A rough schematic of a dissolution cell (containing the filter) for dissolution testing of powders in flow through systems.

**Figure 5 pharmaceutics-15-00983-f005:**
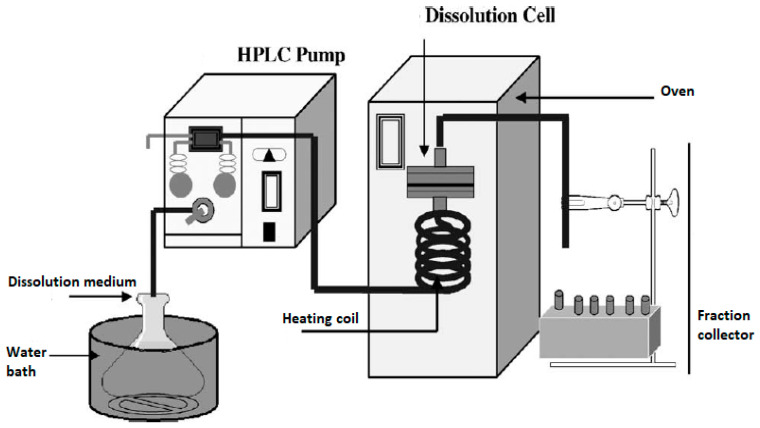
A flow through cell dissolution apparatus (the figure was taken from reference [4]).

**Figure 6 pharmaceutics-15-00983-f006:**
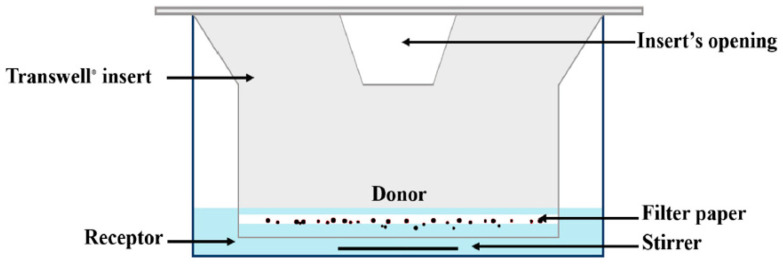
A schematic representation of the Transwell^®^ diffusion cell apparatus (The figure has been taken from reference [18]).

**Figure 7 pharmaceutics-15-00983-f007:**
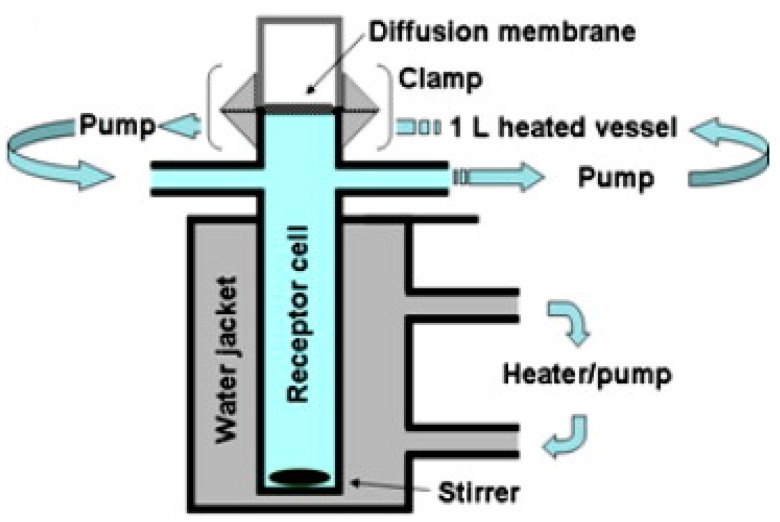
Schematic representation of the Franz diffusion cell (the figure is taken from reference [14]).

**Figure 8 pharmaceutics-15-00983-f008:**
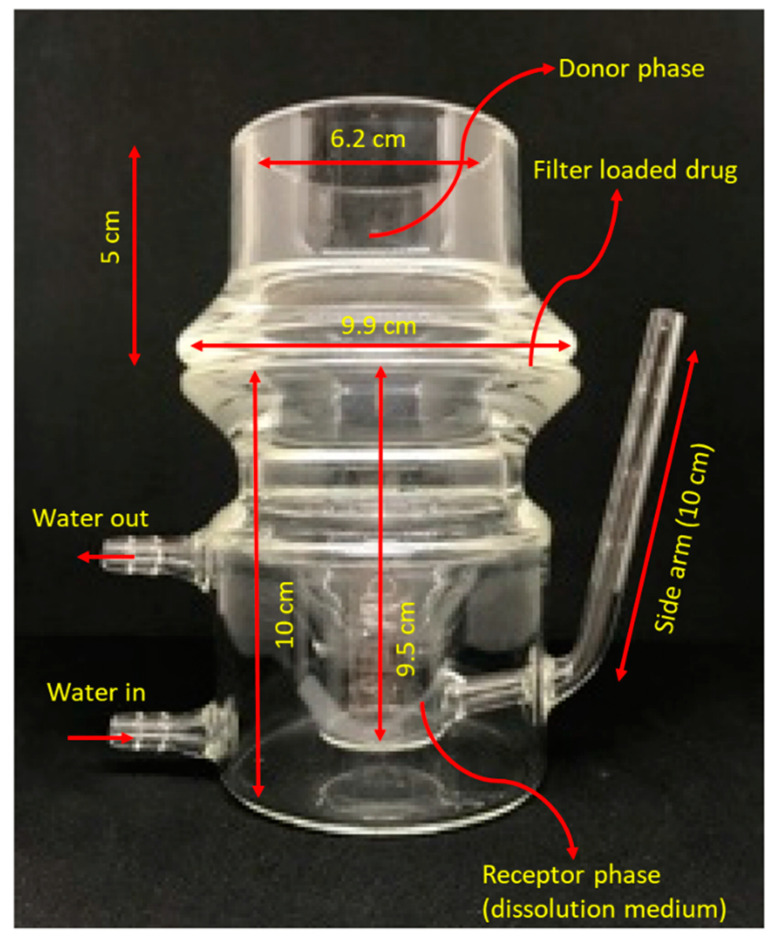
The schematic representation of Respicell (adopted and modified from [22]).

**Figure 9 pharmaceutics-15-00983-f009:**
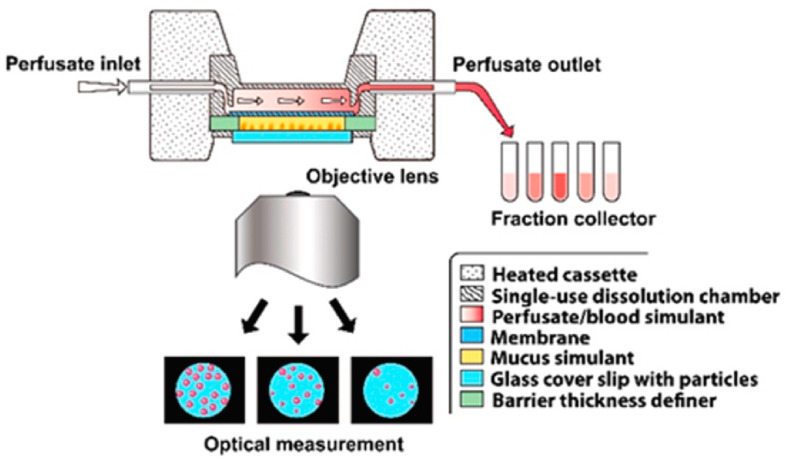
An overview of Dissolvit^®^ (the figure was taken from reference [21]).

**Figure 10 pharmaceutics-15-00983-f010:**
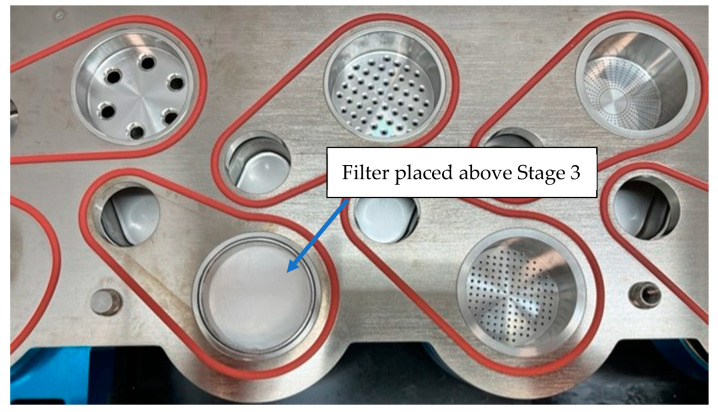
A glass fiber filter is located on stage 3 of the NGI.

**Figure 11 pharmaceutics-15-00983-f011:**
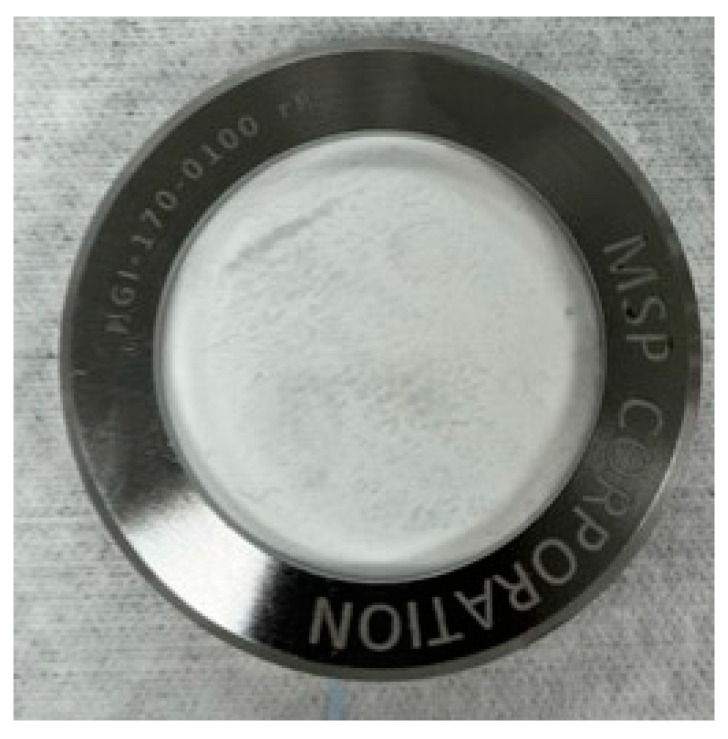
Filter cassette to keep the glass fiber filter and the pre-soaked polycarbonate filter.

**Table 1 pharmaceutics-15-00983-t001:** Dissolution testing methods employed to assess the dissolution of fine particle doses of OIDPs.

Dissolution Apparatus/Permeability Type	Dose Collection Method	Particle Collection Filter/Dissolution Membrane	Rotation Speed or Flow Rate	Reference
United State pharmacopoeia apparatus II (paddle)	Spray drying from aqueous PVA solutions	None	50 rpm	[14]
	rNGI	Glass fiber filter covered with polycarbonate membrane	50−100 rpm	[15]
	ACI (Andersen cascade impactor)	A regenerated cellulose membrane (0.45 μm) is placed into a membrane holder	140 rpm	[16]
Flow through cell apparatus	ACI	A fiberglass filter between 0.45 µm membrane filters in stainless steel holder	0.4−1.5 mL/min	[4]
	ACI (Andersen cascade impactor)	Filter membrane consisting of regenerated cellulose covered with a second membrane	1 mL/min	[16]
	Spray drying from aqueous PVA solutions	A nitrocellulose membrane with a pore size of 0.45 μm is placed between a second membrane filter and a metal mesh screen	0.5 mL/min	[14]
Transwell^®^ system apparatus	ACI	Polyvinylidene difluoride (PVDF) membrane filter with a pore size of 0.22 μm	N/A	[17]
	NGI	Glass microfilter paper (GF/C^TM^)	N/A	[18]
	ACI	Sedimentation onto filter	N/A	[19]
Franz cell apparatus	Spray drying from aqueous PVA solutions	Nitrocellulose membrane with a pore size of 0.45 μm	N/A	[14,20]
	ACI (Andersen cascade impactor)	A regenerated cellulose membrane with a pore size of 0.45 μm	100 rpm	[17]
Dissolvit^®^	PreciseInhale exposure system	Glass coverslip	0.42 mL/min	[21]
Respicell^TM^	fast screening impactor (FSI)	Filter from FSI collects particles less than 5 µm	magnetically stirred	[22]
Cell-based methods	Aerosolization using a PennCentury™	Broncho-epithelial cell line Calu-3 mounted onto Transwells^®^	N/A	[23]
	TSI	Broncho-epithelial cell line Calu-3 mounted onto Transwells^®^	N/A	[24]

## Data Availability

Not applicable.

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
