# Peer review of "In Vitro Dissolution and Permeability Testing of Inhalation Products: Challenges and Advances"

_pharmaceutics, 2023, doi:10.3390/pharmaceutics15030983_

Round 1

Reviewer 1 Report

    Ali et al. summarized the recent advances in the dissolution of inhalation products and their challenges. This review was well-written and summarized. They first listed the parameters affecting the API dissolution of inhaled products and gave a process overview of dissolution testing. Then they introduced the current dissolution test methods in detail and discuss the limitations. They also discussed the challenges in dose collection, and optimization of dissolution medium. Finally, they listed the modeling and comparing profiles analysis to dissolution data. Hence, in my opinion, this review is complete and meaningful. However, there still are some concerns before publication.

Please address below concerns:

1.     Please add the full name in Figure 2 legends for rNGI ACI TSI FSI.

2.     Line 124, the title should be: 2. Process overview…, not 2.2.

3.     Figures 5 and 6, 7,9, is it OK to directly take from reference? It’s inappropriate. Please either get permission from the authors or publisher, or make by yourself.

4.  Line 563, ICS/LABA add the full name when first shown in the text. Please check other abbreviations as well

5.     Line 615 font size is not right.

6.     Line 724, the literature footnote marker is missing “[“. There are other same problems like lines 302, 286, etc. Please check all.

7.     Line 805-848, has different font size as well. Actually, A lot of format/spacing problems of paragraphs in the whole manuscript, please correct this.

8.     Is it necessary to list 2 examples in part 7 with one-page description? 

9.     Might be better to merge parts 9 and 10.

Reviewer 2 Report

The submitted review by Nokhodchi et al. is a very useful and practical summary and advantages/disadvantages of currently available dissolution systems.

Comments

Please check the statement in section 3.5 that A549 are used for permeability testing.

l.568-586: it would be good to have an information on the result.

Section 9 and 10 should be combined to avoid repetition of similar information and because the section “Future perspectives” also contains conclusions.

Minor

l.340: correct “France-cell”
